# Graph Wave Networks

## ABSTRACT

Dynamics modeling has been introduced as a novel paradigm in message passing (MP) of graph neural networks (GNNs). Existing methods consider MP between nodes as a *heat diffusion* process, and leverage *heat equation* to model the temporal evolution of nodes in the embedding space. However, heat equation can hardly depict the wave nature of graph signals in graph signal processing. Besides, heat equation is essentially a partial differential equation (PDE) involving a first partial derivative of time, whose numerical solution usually has low stability, and leads to inefficient model training. In this paper, we would like to depict more wave details in MP, since graph signals are essentially wave signals that can be seen as a superposition of a series of waves in the form of eigenvector. This motivates us to consider MP as a *wave propagation process* to capture the temporal evolution of wave signals in the space. Based on *wave equation* in physics, we innovatively develop a *graph wave equation* to leverage the wave propagation on graphs. In details, we demonstrate that the graph wave equation can be connected to traditional spectral GNNs, facilitating the design of *graph wave networks (GWNs)* based on various Laplacians and enhancing the performance of the spectral GNNs. Besides, the graph wave equation is particularly a PDE involving a second partial derivative of time, which has stronger stability on graphs than the heat equation that involves a first partial derivative of time. Additionally, we theoretically prove that the numerical solution derived from the graph wave equation are *constantly stable*, enabling to significantly enhance model efficiency while ensuring its performance. Extensive experiments show that GWNs achieve state-of-the-art and efficient performance on benchmark datasets, and exhibit outstanding performance in addressing challenging graph problems, such as over-smoothing and heterophily. Our code is available at https://anonymous.4open.science/r/GWN/.

## CCS CONCEPTS

• **Computing methodologies → Neural networks**.

## KEYWORDS

Graph Neural Networks, Partial Differential Equations, Wave Equation

## 1 INTRODUCTION

The widespread availability of graph data has propelled the development of *graph neural networks (GNNs)* [46]. These methods have achieved significant success in various applications, such as recommendation systems [45], social network analysis [26], particle physics [36], and drug discovery [13].

Currently, mainstream GNNs [10, 19] develop *message passing (MP)* paradigm with graph signal processing [37], which delivers messages node-by-node by stacking graph convolutional layers. In fact, the MP paradigm can be modelled as a differential equation [6]. Recent studies explore *partial differential equations (PDEs)* [6, 23, 40] to consider the spatial and temporal relationships in MP. Intuitively,

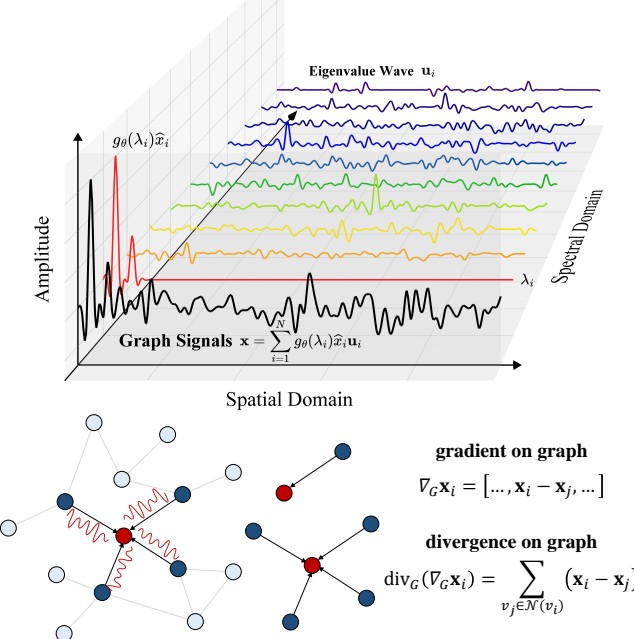

**Figure 1: Top: the partial spectrum on the real-world dataset Cora, where the graph signal can be treated as a superposition of multiple eigenvector waves $\mathbf{u}_i$ with amplitude $g_\theta(\lambda_i)\widehat{x}_i$ (detailed in Eq. (7)). Bottom: the mechanism of wave propagation on the graph (detailed in Sec. 4.2).**

they analogize MP to a *heat diffusion* process between nodes, and thus leverage the heat equation to model nodes in the embedding space. In physics, the heat equation describes the evolution of temperature in the space over time. Consider 3-dimension spatial variables $(\omega_1, \omega_2, \omega_3)$ in the space and a time variable $t$, the **Heat Equation** is:

$$\frac{\partial u}{\partial t} = \alpha \left( \frac{\partial^2 u}{\partial \omega_1^2} + \frac{\partial^2 u}{\partial \omega_2^2} + \frac{\partial^2 u}{\partial \omega_3^2} \right) = \alpha \cdot \text{div}(\nabla u), \quad (1)$$

where $u$ is the abbreviation symbol for $u(\omega_1, \omega_2, \omega_3, t)$, denoting the temperature at position $(\omega_1, \omega_2, \omega_3)$ and time $t$, and $\alpha$ is a coefficient called the thermal diffusivity of the medium. However, in context of graph learning, these methods suffer from two drawbacks: *First, the heat equation can hardly depict the wave nature of graph signals.* In graph signal processing theory [37], the graph can be seen as a combination of waves with different frequencies, where we show the graph wave spectrum in Figure 1 (Top). The heat equation is naturally not capable to process the details of wave property in message passing, which leads to coarse and sub-optimal GNN designs. *Second, the heat equation is essentially a PDE involving a first partial derivative of time, yet its stability of numerical solution can be poor on graph.* This requires small time step lengths in solving

**Table 1: Graph heat equation and graph wave equation. X = X(t), ∇ and div denote gradient and divergence.**

| Graph Heat Equation | $\frac{\partial \mathbf{X}}{\partial t} = \text{div}\left[\text{diag}\left(a(\mathbf{x}_i, \mathbf{x}_j)\right)\nabla \mathbf{X}\right], \mathbf{X} \in \mathbb{R}^{N \times d}$ | |
|---|---|---|
| | $(\nabla \mathbf{X})_{ij} = \mathbf{x}_j - \mathbf{x}_i$ | $\text{div}[\nabla \mathbf{X}]_i = \sum_{v_j \in N(v_i)} (\mathbf{x}_j - \mathbf{x}_i)$ |
| Graph Wave Equation | $\frac{\partial \mathbf{X}}{\partial t} = a^2[\dots, \text{div}_G((\nabla_G \mathbf{x})_i), \dots]^\top, \mathbf{X} \in \mathbb{R}^{N \times d}$ | |
| | $\nabla_G \mathbf{x}_i = [\dots, \mathbf{x}_i - \mathbf{x}_j, \dots]$ | $\text{div}_G(\nabla_G \mathbf{x}_i) = \sum_{v_j \in N(v_i)} (\mathbf{x}_i - \mathbf{x}_j)$ |

PDE, leading to low efficiency in model training, and can be easily affected by initial values and generate unsatisfactory performance. We also give the experimental evidence in Sec. 5.

Unlike previous studies [6, 23, 40] that regard MP as a heat diffusion process, we innovatively analogize MP to a *wave propagation* process. Particularly, MP basically embodies the information propagation along the nodes at different spatial positions and temporal steps, which bears a strong resemblance to the propagation of *waves* in physics, such as electromagnetic waves. To depict the wave propagation process, we explore the wave equation, a PDE involving the second derivative of time, which describes the evolution of wave intensity over time and has wide applicability in physics. Also consider the three spatial variables $(\omega_1, \omega_2, \omega_3)$ and a time variable $t$, the **Wave Equation** is:

$$\frac{\partial^2 u}{\partial t^2} = a^2 \left( \frac{\partial^2 u}{\partial \omega_1^2} + \frac{\partial^2 u}{\partial \omega_2^2} + \frac{\partial^2 u}{\partial \omega_3^2} \right) = a^2 \cdot \text{div}(\nabla u), \quad (2)$$

where $u$ is the abbreviation symbol for $u(\omega_1, \omega_2, \omega_3, t)$, denoting the wave intensity at position $(\omega_1, \omega_2, \omega_3)$ and time $t$, and $a$ is a coefficient denoting the propagation speed of the wave. Notably, though the wave equation has similar formulation as heat equation, it has intrinsically different properties in the context of graph learning: *First, wave equation is naturally suitable for MP in GNNs*. Since GNN is actually a wave filter of frequencies, it can be achieved by Laplacian. It investigates more details in processcing graph signals, offering higher accuracy and practicality compared to heat equation. *Second, the wave equation is essentially a PDE involving a second partial derivative of time, which can offer stronger stable condition on graph.* This property provides more efficient training process that allows larger time step lengths, and often generates robust experimental results (see Figure 4). The wave propagation process on graph can be depicted as Figure 1 (Bottom).

Though applying the above wave equation to MP is attractive and reasonable, the specific MP formulation of wave equation is not obvious, and it is non-trivial to conduct further exploration. To this end, our breakthrough point is to *formulate the MP process with the wave equation at each node*, since once the MP process is specified, the overall GNN is designed. Therefore, we first would like to derive graph wave equation for MP. Inspired by Chamberlain et al. [6], we let the gradient on graph be the difference of features between a central node and each of its neighbors, and the divergence be the total difference of them. Thereby we can derive the formulation in *graph wave equation*, and the details are shown in Table 1. By introducing the characteristics of graph Laplacian, the graph wave

equation can be further rewritten into a brief form for the solution of PDEs (detailed in Sec. 4.2).

Based upon the above derived graph wave equation, we then would like to achieve the MP for GNN design by solving PDEs. In fact, the solution of PDEs is often an iterative node feature update process, which can actually be interpreted as the MP process. Existing PDE-based GNN methods [6, 29] utilizing the *forward Euler method* to solve PDEs, resulting in conditionally stable *explicit schemes*, making it difficult to balance the performance and efficiency of the model. Besides, Chamberlain et al. [6] also attempt *implicit schemes* through the *backward Euler method*, which are constantly stable but require solving linear systems, leading to higher computational complexity and lower efficiency compared to the explicit schemes. In this paper, we use the *forward Euler method* to solve the graph wave equation, which can obtain **constantly stable explicit schemes**. This allows the model to significantly improve convergence rates by selecting larger time step lengths, thereby enhancing efficiency of operation while guaranteeing model performance (detailed in Sec. 4.3). Based upon the above designs, we propose the Graph Wave Networks (termed as GWN) with MP formulated by graph wave equation. Two specified implementations are provided to achieve our GWN according to different specification of Laplacians (detailed in Sec. 4.4).

Our main contributions can be summarized as follows:

- For the first time, we model the message passing from an innovative perspective of a **wave propagation process**, thereby maintaining the wave nature in graph convolutional operation.
- We develop wave equation into **graph wave equation**, and propose a novel **graph wave network**, namely GWN. It establishes a connection between wave equation and traditional spectral GNNs, enhancing them with more details of waves on graphs.
- Through the theoretical evidence, we prove that the explicit scheme of the graph wave equation is **constantly stable**, allowing **significant efficiency improvements** while guaranteeing robust model prediction results.
- We conduct extensive experiments on 9 benchmark datasets. Experimental results substantiate that GWN outperforms previous state-of-the-art methods and achieves efficient performance in alleviating over-smoothing and heterophily.

## 2 RELATED WORK

*Spectral GNNs.* Spectral GNNs [5] based on Laplacian to design various filter functions in the spectral domain. One category of spectral GNNs directly modify the Laplacian. GCN [19] incorporates self-loops in the normalized Laplacian, which manifests as a low-pass filter. Some methods [3, 14, 28, 53] introduce additional high-pass filters to learn difference between nodes. Bianchi et al. [2] propose an auto-regressive moving average filter to capture the global graph structure. Defferrard et al. [10] and He et al. [17] approximate the spectral graph convolution using Chebyshev polynomial. He et al. [16] approximate arbitrary filters using Bernstein polynomials. Wang and Zhang [42] utilize Jacobi polynomials to adapt a wider range of weight functions. Chien et al. [9] employ learnable parameters to approximate the polynomial coefficients.

These methods are all built upon the traditional graph signal processing methods, but few of them fully explore the nature of wave propagation in the MP process.

*Differential Equations on Graphs.* Neural ODE [8] models the embedding representation as a continuous dynamic with respect to neural network parameters. Poli et al. [31] propose a continuous-depth GNN framework and solves the forward process using numerical methods. Xhonneux et al. [47] establish the relationship between the derivative of node embeddings and the initial and neighboring embeddings of nodes. Rusch et al. [34] relate to traditional GNNs using second-order ODEs. Nguyen et al. [29] adapt well-known Kuramoto model to alleviate over-smoothing. Neural PDE [24] applies partial differential equations to graph-based problems. Eliasof et al. [11] utilizes non-linear diffusion and non-linear hyperbolic equations to model message passing. Wang et al. [43] decouple the terminal time and feature propagation steps from a continuous perspective. Klicpera et al. [20] define graph diffusion convolution to overcome the limitation of traditional GNNs in aggregating only direct neighbors. Zhao et al. [52] propose adaptive diffusion convolution to automatically learn the optimal neighborhood from the data. Recent works introduce the heat diffusion equation on graphs to simulate the temporal dynamics of node embeddings [6]. The explicit scheme stability derived from GRAND is conditional, ensuring model performance only when using smaller time steps, leading to inefficient model performance. Thorpe et al. [40] further utilize a heat diffusion equation with a source term to define graph convolution, which performs better in low-label-rate scenarios. Li et al. [23] introduce a general diffusion equation framework with a fidelity term and establishes the connection between the diffusion process and GNNs. Compared to considering message passing as a heat diffusion process between nodes, treating it as a wave propagation process better aligns with the process of inter-node information interaction described in graph signal processing. Moreover, note that there are several DE-based GNNs [11, 23, 40, 43, 47] lack stability proofs, which can hardly ensure the robustness of models, and weaker stability conditions make it challenging to balance model performance and efficiency. *In contrast, in this paper, the proposed graph wave networks can significantly enhance efficiency while ensuring model performance, due to its constantly stable properties.*

## 3 PRELIMINARIES

In this section, we discuss graph signal processing in traditional GNNs, and then introduce fundamental concepts of wave equations.

### 3.1 Graph Signal Processing

Graph signal processing [37] is a field that focuses on signal analysis and processing conducted on graphs.

**Notations of Graph.** Given a simple undirected graph $\mathcal{G}$, composed of a set of nodes $\mathcal{V}$ and a set of edges $\mathcal{E}$, with $N = |\mathcal{V}|$ nodes in total. Graph $\mathcal{G}$ is associated with a feature matrix $\mathbf{X} \in \mathbb{R}^{N \times d}$, where the $i$-th row of $\mathbf{X}$ corresponds to the feature vector $\mathbf{x}_i = [x_{i1}, x_{i2}, \ldots, x_{id}] \in \mathbb{R}^d$ of node $v_i$, and the $j$-th column of $\mathbf{X}$ corresponds to the graph signal of dimension $j$. We denote the adjacency matrix as $\mathbf{A}$ and the degree matrix as $\mathbf{D}$ with $D_{ii} = \sum_i A_{ij}$.

**Spectral Graph Convolution.** Traditional spectral GNNs design spectral graph convolution using the symmetric normalized Laplacian $\mathbf{L}_{sym} = \mathbf{I} - \mathbf{D}^{-1/2}\mathbf{A}\mathbf{D}^{-1/2}$, which can be decomposed into $\mathbf{L}_{sym} = \mathbf{U}\mathbf{\Lambda}\mathbf{U}^\top$, where $\mathbf{\Lambda} = \text{diag}(\lambda_1, \lambda_2, \ldots, \lambda_N)$ is a diagonal matrix of eigenvalues and $\mathbf{U}$ is a matrix of corresponding eigenvectors. Here, the eigenvalues satisfy $0 = \lambda_1 \leq \cdots \leq \lambda_N = 2$. Based on this, the definition of spectral graph convolution is as follows:

$$(\mathbf{g} * \mathbf{X})_G = \mathbf{U}\mathbf{g}_\theta\mathbf{U}^\top\mathbf{X}, \tag{3}$$

where $\mathbf{g}_\theta = \mathbf{g}_\theta(\mathbf{\Lambda}) = \text{diag}(g_\theta(\lambda_1), \ldots, g_\theta(\lambda_N))$ denotes the graph filter function.

### 3.2 Wave Equation

The wave equation describes the laws of waves in the space evolving over time, which has been widely adopted to analyze the characteristics of waves like electromagnetic waves in physics.

Given that $u(\omega_1, \ldots, \omega_d, t)$ is a scalar-valued function on $\Omega^d \times [0, \infty)$ that reflects the wave intensity, where $\Omega^d = \omega_1 \times \cdots \times \omega_d$ denotes the spatial dimension, and $t \in [0, \infty)$ denotes the time dimension, the wave equation can be formulated by the following partial differential equation (PDE):

$$\frac{\partial^2 u}{\partial t^2} = a^2 \left( \frac{\partial^2 u}{\partial \omega_1^2} + \cdots + \frac{\partial^2 u}{\partial \omega_d^2} \right) = a^2 \Delta u = a^2 \cdot \text{div}(\nabla u), \tag{4}$$

where $\Delta = \sum_i^d \frac{\partial^2}{\partial \omega_i^2}$ denotes Laplacian operator in mathematics, $\nabla$ and div denote gradient and divergence, respectively. $a$ denotes the propagation velocity of waves in the medium (such as the propagation velocity of electromagnetic waves in vacuum), which is a scalar or a function to describe the wave.

## 4 WAVE EQUATION ON GRAPHS

In this section, we would like to analyze the wave nature of graph signals, and propose graph wave equation with solutions for MP.

### 4.1 Graph and Wave

This section illustrates the connection between graphs and waves from the perspective of graph signal processing. For convenience, we use a toy example of graph signals $\mathbf{x} = [x_1, \ldots, x_N]^\top \in \mathbb{R}^{1 \times N}$ with $N$ nodes embedded in 1-dimensional space $\Omega^1$, where the same principle can be extended to $d$-dimensional space $\Omega^d$. First, the typical implementation of spectral graph convolution transforms graph signals by Fourier transform:

$$\widehat{\mathbf{x}} = \mathbf{U}^\top\mathbf{x} = \left[ \sum_{j=1}^N u_{1j}x_j, \ldots, \sum_{j=1}^N u_{Nj}x_j \right]^\top \in \mathbb{R}^N, \tag{5}$$

where the $i$-th element $\widehat{x}_i = \sum_{j=1}^N u_{ij}x_j = <\mathbf{u}_i, \mathbf{x}>$ of $\widehat{\mathbf{x}}$ denotes the signal strength of the spectral signal corresponding to the eigenvalue $\lambda_i$ in the spectral domain. Then, based upon them, the spectral graph convolution is defined as:

$$\mathbf{g}_\theta\mathbf{U}^\top\mathbf{x} = [g_\theta(\lambda_1)\widehat{x}_1, \ldots, g_\theta(\lambda_N)\widehat{x}_N]^\top \in \mathbb{R}^N. \tag{6}$$

Finally, the inverse Fourier transform is used to map the spectral signal back to the spatial domain:

$$\mathbf{L}_{sym}\mathbf{x} = \mathbf{U}\mathbf{g}_\theta\mathbf{U}^\top\mathbf{x} = [\mathbf{u}_1, \ldots, \mathbf{u}_N]\,[g_\theta(\lambda_1)\widehat{x}_1, \ldots, g_\theta(\lambda_N)\widehat{x}_N]^\top$$

$$= \sum_{i=1}^{N} g_\theta(\lambda_i)\widehat{x}_i\mathbf{u}_i \in \mathbb{R}^N. \tag{7}$$

**Remark 1.** *The connection between graph and wave.* From Eq. (7), we can observe that the graph signal in *spatial* domain can be treated as a linear combination of $N$ different eigenvalues in *spectral* domain, vividly depicted in Figure 1 (Top). Intuitively, we can interpret the eigenvalue $\lambda_i$ as the frequency, its corresponding eigenvector $\mathbf{u}_i$ as a particular type of wave, and $g_\theta(\lambda_i)\widehat{x}_i$ denotes the amplitude of the wave. This reflects the wave nature of the spectral graph convolution on any given graphs.

## 4.2 Graph Wave Equation

We are going to extend the wave equation on the graph. Given the graph signal $\mathbf{X} \in \mathbb{R}^{N\times d}$ (*i.e.*, node features), for any node $v_i$, its node representation acquires messages from its neighbors $v_j \in \mathcal{N}(v_i)$ in GNNs. Then, we can derive the gradient of $v_i$ in MP on its graph $G$ by vector subtraction between $v_i$ and its neighbors, which actually captures their signal differences [6]:

$$\nabla_G\mathbf{x}_i := [\ldots, \mathbf{x}_i - \mathbf{x}_j, \ldots]^\top \in \mathbb{R}^{|\mathcal{N}(v_i)|\times d}, \tag{8}$$

where the gradient direction reflects the direction from $v_j$ to $v_i$ in MP, and the gradient magnitude measures the feature difference amount between $v_i$ and $v_j$. Subsequently, we can derive the divergence of $v_i$ on $G$ by the sum of feature differences between the node $v_i$ and all its neighbor $v_j \in \mathcal{N}(v_i)$:

$$\text{div}_G(\nabla_G\mathbf{x}_i) := \sum_{v_j \in \mathcal{N}(v_i)} (\mathbf{x}_i - \mathbf{x}_j) \in \mathbb{R}^d, \tag{9}$$

where the divergence actually measures the total difference between a node and its neighborhood. Finally, by substituting Eq. (9) into (4), we propose **Graph Wave Equation** on the graph:

$$\frac{\partial^2\mathbf{X}}{\partial t^2} = a^2\,[\ldots, \text{div}_G(\nabla_G\mathbf{x}_i), \ldots]^\top \in \mathbb{R}^{N\times d}. \tag{10}$$

where $\mathbf{X}$ is the signal intensity (of all nodes) in the entire graph $G$, and can be regarded as the node representations. Each row of $\mathbf{X}$ involves a wave equation at a specific node. By introducing the form of Laplacian, we can rewrite Eq. (10) as follows (*see Appendix A.1 for mathematical proof*):

**Proposition 1.** Let $\mathbf{L} = \mathbf{D} - \mathbf{A}$ denote the discrete form of the Laplacian operator $\Delta$. The *graph wave equation* can be rewritten as:

$$\frac{\partial^2\mathbf{X}}{\partial t^2} = a^2\mathbf{L}\mathbf{X} := \mathbf{L}_a\mathbf{X}. \tag{11}$$

where we incorporate $a$ into $\mathbf{L}$ for a simple form, *i.e.*, $\mathbf{L}_a := a^2\mathbf{L}$. Here, the propagation velocity $a$ controls the speed of wave propagation between nodes on the graph. Note that the propagation velocity $a$ can alternatively be constant or learnable parameters in practice, leading to different forms of Laplacian (detailed in Eq. (15) and (17) later). Benefit by treating $a$ as learnable parameters, we can redefine Laplacian with flexibility and establish a connection between wave equation and Laplacian-based spectral GNNs.

**Remark 2.** *The connection between graph wave equation and spectral GNNs.* With Eq. (11), we can easily combine the wave equation with spectral GNNs into graph wave equations. Thereby, the graph wave equation can be flexibly extended and achieved by specific designed Laplacians of existing spectral GNNs.

## 4.3 Solution of Graph Wave Equation

In general, it is often intractable to obtain an analytical solution for a PDE. Fortunately, there are numerical methods that can approximate PDE solutions. First, the continuous PDE can be discretized into a finite form of a linear algebraic system using the finite difference method. Then, an iteration process with initial values can be employed to solve this linear algebraic system. In the context of graphs, only the time dimension needs to be further discretized on each node. To this end, we employ a commonly-used discretization method in mathematics, namely *forward Euler method*, to solve the graph wave equation for node representations (i.e., $\mathbf{X}$) [6].

Formally, the forward Euler method discretizes the graph wave equation by performing forward difference in the time dimension, and then derives the **explicit scheme**:

$$\frac{\mathbf{X}^{(n+1)} - 2\mathbf{X}^{(n)} + \mathbf{X}^{(n-1)}}{\tau^2} = \mathbf{L}_a^{(n)}\mathbf{X}^{(n)}, \tag{12}$$

where $\tau$ is the time step length. The initial values of $\mathbf{X}^{(0)}$ and $\mathbf{X}^{(1)}$ in the PDE can be obtained using the second-order central quotient:

$$\mathbf{X}^{(0)} = \varphi_0(\mathbf{X}), \quad \mathbf{X}^{(1)} = \tau\varphi_1(\mathbf{X}) + \left(\mathbf{I} + \frac{\tau^2}{2}\mathbf{L}_a^{(0)}\right)\varphi_0(\mathbf{X}), \tag{13}$$

where $\varphi_0(\mathbf{X})$ and $\varphi_1(\mathbf{X})$ can be practically achieved by neural networks, such as feedforward networks. Finally, the explicit scheme of the graph wave equation is given by:

$$\mathbf{X}^{(n+1)} = \left(2\mathbf{I} + \tau^2\mathbf{L}_a^{(n)}\right)\mathbf{X}^{(n)} - \mathbf{X}^{(n-1)}. \tag{14}$$

*Please see Appendix A.2 for the detailed derivation of the forward Euler method.* In particular, we can interpret the above equation from the perspective of message passing:

**Remark 3.** *The perspective of message passing.* By decomposing $\mathbf{X}^{(n+1)}$ into $\mathbf{X}^{(n+1)} = \left(\mathbf{I} + \tau^2\mathbf{L}_a^{(n)}\right)\mathbf{X}^{(n)} + \left(\mathbf{X}^{(n)} - \mathbf{X}^{(n-1)}\right)$, the graph signal at time $t_{n+1}$ consists of two components: 1) the aggregation of neighbor information at time $t_n$, and 2) the difference between the graph signal at times $t_n$ and $t_{n-1}$.

## 4.4 Graph Wave Networks

In this section, we propose two specifications of GWNs with time-independent and time-dependent Laplacian, respectively.

*Symmetric Normalized Laplacian.* We consider time-independent Laplacian, where we let the velocity be a constant. Typically, we adopt GCN [19] as the base model and design a symmetric normalized Laplacian without self-loops, which behaves as a low-pass filter in the spectral domain:

$$\mathbf{L}_a = \mathbf{D}^{-\frac{1}{2}}\mathbf{A}\mathbf{D}^{-\frac{1}{2}}. \tag{15}$$

**GWN-sym.** We propose the Graph Wave Network based on the symmetric normalized Laplacian. With the initial values given by

Eq. (13), the feature matrix at time $t_{n+1}$ can be determined as:

$$\mathbf{X}^{(n+1)} = \left(2\mathbf{I} + \tau^2 \mathbf{D}^{-\frac{1}{2}} \mathbf{A} \mathbf{D}^{-\frac{1}{2}}\right) \mathbf{X}^{(n)} - \mathbf{X}^{(n-1)}. \qquad (16)$$

The final feature matrix at the terminal time $T$ is transformed into a prediction matrix by an MLP as $\mathbf{Y} = \text{MLP}(\mathbf{X}^{(T)})$.

*Frequency Adaptive Laplacian.* Next, we consider time-dependent Laplacian, where the velocity is non-constant learnable parameters. Typically, we adopt FAGCN [3] as the base model and propose a frequency adaptive Laplacian with a time-dependent learnable parameter $\varepsilon^{(n)} \in (0, 1)$, which designs both a low-pass filter and a high-pass filter in the spectral domain:

$$\mathbf{L}_{a,l}^{(n)} = \varepsilon^{(n)} \mathbf{I} + \mathbf{D}^{-\frac{1}{2}} \mathbf{A} \mathbf{D}^{-\frac{1}{2}}, \ \mathbf{L}_{a,h}^{(n)} = \varepsilon^{(n)} \mathbf{I} - \mathbf{D}^{-\frac{1}{2}} \mathbf{A} \mathbf{D}^{-\frac{1}{2}}. \qquad (17)$$

**GWN-fa.** We propose the G̲raph W̲ave N̲etwork based on the frequency a̲daptive Laplacian. The initial values are also given by Eq. (13), and the feature matrix at time $t_{n+1}$ can be determined as (*see Appendix A.4 for detailed derivation*):

$$\mathbf{X}^{(n+1)} = \varepsilon^{(n)} \mathbf{X}^{(0)} + \left(2\mathbf{I} + \tau^2 \boldsymbol{\alpha}^{(n)} \odot \mathbf{D}^{-\frac{1}{2}} \mathbf{A} \mathbf{D}^{-\frac{1}{2}}\right) \mathbf{X}^{(n)} - \mathbf{X}^{(n-1)}. \qquad (18)$$

where the element $\boldsymbol{\alpha}_{ij}^{(n)} = \tanh(\mathbf{g}^{(n)\top}[\mathbf{x}_i^{(n)} || \mathbf{x}_j^{(n)}])$ of $\boldsymbol{\alpha}^{(n)}$ is the attention weight between nodes $v_i$ and $v_j$ at time $t_n$, and $\mathbf{g}^{(n)} \in \mathbb{R}^{2d}$ is a learnable parameter vector. Notably, when $\alpha_{ij}^{(n)} > 0$, the two nodes are more similar and GWN-fa behaves a low-pass filter; and when $\alpha_{ij}^{(n)} < 0$, two nodes are more dissimilar and GWN-fa behaves a high-pass filter. The final feature matrix at terminal time $T$ is also transformed into a prediction matrix by an MLP.

### 4.5 Stability

Stability is an important property of differential equations, which is closely related to the robustness in machine learning [6], and refers to the property that small perturbations in initial values would not result in a significant change in solutions. We discuss the initial value stability of the explicit scheme $\mathbf{U}^{(k+1)} = \mathbf{C}\mathbf{U}^{(k)}$. Formally, if there exists $\tau_0 > 0$ and a constant $K > 0$ such that the inequality $\|\mathbf{U}^{(k+1)}\| = \|\mathbf{C}^{k+1}\mathbf{U}^{(0)}\| \leq K\|\mathbf{U}^{(0)}\|$ holds for all $0 < \tau \leq \tau_0$ and $0 < k\tau \leq T$, then the explicit scheme is said to be initial value stable. It is crucial to prove the stability of explicit schemes to ensure that the resulting GNNs is reliable and feasible. A commonly used method for proving stability is the matrix method:

**Theorem 1.** Let $\rho(\mathbf{C}) = |\lambda|_{max}$ denote the spectral radius of matrix C, if $\rho(\mathbf{C}) \leq 1$, the numerical scheme is stable.

Based on Theorem 1, we prove that both the explicit schemes based on the symmetric normalized Laplacian (*see Appendix A.3 for proof*) and the frequency adaptive Laplacian (*see Appendix A.5 for proof*) are **constantly stable**.

**Theorem 2.** Given $\mathbf{L}_a = \mathbf{D}^{-1/2}\mathbf{A}\mathbf{D}^{-1/2}$ with $\lambda \in [-1, 1]$, the explicit scheme is constantly stable for any $\tau \in R^+$.

**Theorem 3.** Given $\mathbf{L}_{a,\cdot} = \varepsilon\mathbf{I} \pm \mathbf{D}^{-\frac{1}{2}}\mathbf{A}\mathbf{D}^{-\frac{1}{2}}$ with $\lambda \in [\varepsilon - 1, \varepsilon + 1]$, the explicit scheme is constantly stable for any $\tau \in R^+$.

**Remark 4.** *The impact of constantly stability for models.* Theoretically, we prove that the explicit scheme of the graph wave equation is constantly stable, guaranteeing the model performance would not be affected by the time step length, thus allowing to enhance

the convergence rate and significantly improve the efficiency by choosing a relatively larger $\tau$.

*Comparison with heat equation based GRAND.* The explicit scheme of GRAND is $\mathbf{X}^{(n+1)} = \left(\mathbf{I} + \tau\overline{\mathbf{A}}\left(\mathbf{X}^{(n)}\right)\right)\mathbf{X}^{(n)}$, where $\overline{\mathbf{A}}\left(\mathbf{X}^{(n)}\right)$ is an attention matrix constrained to be a right stochastic matrix satisfying $\sum_{j=1}^N \alpha_{ij} = 1$ and $\alpha_{ij} > 0$. Benefiting from being a right stochastic matrix, the explicit scheme is stable and can be directly proven. However, it has the two deficiencies: First, the stability of explicit schemes is conditional (*i.e.*, $0 < \tau < 1$), striking a balance between performance and efficiency. Detailed comparisons and validations can be found in Sec. 5.4. Second, the attention scores are constrained to $\alpha_{ij} > 0$, which performs poorly in distinguishing inter-class nodes. In contrast, our GWN-fa acts as low-pass and high-pass filters when $\alpha_{ij} > 0$ and $\alpha_{ij} < 0$ respectively, enabling better handling of various types of graph.

### 4.6 Complexity Analysis

The number of learnable parameters of GWN-sym and GWN-fa are $2fd + dc$ and $2fd + (2d+1)T/\tau + dc$, compared to $fd + 2d^2 + dc$ in GRAND [6]. Here, $f$, $d$ and $c$ denote the input dimension, the hidden layer dimension, and the number of classes, respectively. In general, $T/\tau < d$. The computational complexity of each layer of GWN-sym and GWN-fa are $O(Md)$ and $O((N + M)d)$, compared to $O(M'd)$ in GRAND. Here, $N$, $M$ and $M'$ are the number of nodes, edges and rewritten edges.

## 5 EXPERIMENTS

### 5.1 Experimental Setup

**Datasets.** Following the practices [6, 34], we conduct node classification [35] on the following real-world datasets (*see Appendix B.1 for statistics*): (1) *homophilic datasets* [35, 50]: citation networks Cora, CiteSeer and PubMed, Amazon co-purchase networks Computers and Photo, co-author networks CS; (2) *heterophilic datasets* [30, 32, 33]: WebKB datasets Texas, Cornell and Wisconsin.

**Baselines.** We categorize all baselines into the following two classes: (1) *mainstream GNNs*: GCN [19], GAT [41], GraphSAGE [15], SGC [44], JK-Net [48], ResGCN [21], GCNII [7], FAGCN [3], GPR-GNN [9], AIR [51], MixHop [1], Geom-GCN [30], $H_2$GCN [54], LINKX [25], WRGAT [39], GGCN [49], NLMLP [27], GloGNN [22], NSD [4], ACM-GCN [28], Ordered GNN [38]; (2) *GNNs based on differential equation*: CGNN [47], GDC [20], ADC [52], GADC [52], GRAND [6], GraphCON [34]. Unless specifically state that the results are from original papers, baselines are reproduced using their open-source code with fair settings.

**Setup.** We split the datasets into training/valiation/test sets using two schemes: 60%/20%/20% [3, 9] and 48%/32%/20% [4, 30, 49, 54]. During the training phase, our method utilizes the cross-entropy loss function, Adam optimizer [18], and early stopping strategy. The code is implemented using the PyTorch Geometric library [12] and parameter search is performed using wandb library. Finally, we report the mean accuracy and standard deviation of 10 runs.

### 5.2 Performance

To validate the feasibility of GWN, we first compare it with GNNs based on differential equations on 6 homophilic datasets. Table 2

**Table 2: The results of homophilic datasets: mean accuracy ± standard deviation on 60%/20%/20% random splits of data and 10 runs. ∗ models use the best variants, − indicate the original paper did not report this result.**

| Datesets | Cora | CiteSeer | PubMed | Computers | Photo | CS |
|---|---|---|---|---|---|---|
| GCN | 87.51 ± 1.38 | 80.59 ± 1.12 | 87.95 ± 0.83 | 86.09 ± 0.61 | 93.04 ± 0.53 | 95.14 ± 0.25 |
| GAT | 87.42 ± 1.46 | 80.16 ± 1.63 | 85.91 ± 1.26 | 87.89 ± 0.82 | 93.53 ± 0.58 | 94.37 ± 0.28 |
| GraphSAGE | 87.50 ± 1.45 | 79.39 ± 1.38 | 89.64 ± 0.73 | 88.53 ± 0.70 | 94.49 ± 0.55 | 95.93 ± 0.25 |
| CGNN | 88.29 ± 1.11 | 79.93 ± 1.04 | 89.46 ± 0.52 | 88.31 ± 0.65 | 94.35 ± 0.56 | 96.21 ± 0.32 |
| GDC | 86.89 ± 1.28 | 80.05 ± 0.60 | 86.18 ± 0.42 | 88.56 ± 0.38 | 93.56 ± 0.33 | 94.82 ± 0.22 |
| ADC∗ | 87.45 ± 0.89 | 79.43 ± 0.96 | 90.23 ± 0.39 | 88.62 ± 0.56 | 95.33 ± 0.27 | 95.82 ± 0.15 |
| GADC | 87.64 ± 0.64 | 78.62 ± 0.57 | 88.58 ± 0.48 | 87.78 ± 0.54 | 94.70 ± 0.35 | 96.16 ± 0.29 |
| GRAND∗ | 88.70 ± 0.99 | 81.56 ± 1.28 | 88.39 ± 0.32 | 89.37 ± 0.41 | **95.79 ± 0.59** | 95.77 ± 0.28 |
| GraphCON∗ | 87.81 ± 0.92 | 79.68 ± 1.23 | 88.54 ± 1.32 | − | − | − |
| **GWN-sym** | 89.61 ± 0.87 | **81.81 ± 1.70** | 90.56 ± 0.54 | 90.10 ± 0.87 | 95.31 ± 0.65 | 96.66 ± 0.26 |
| **GWN-fa** | **89.66 ± 1.29** | 80.89 ± 1.51 | **90.64 ± 0.73** | **90.62 ± 0.61** | 95.61 ± 0.53 | **96.67 ± 0.26** |

**Table 3: The results of heterophilic datasets: mean accuracy ± standard deviation on 10 runs. ∗ models use the best variants, † results of baselines from papers, ‡ results of baselines are reproduced, − indicate the original paper did not report this result.**

| Datesets | Texas | | Cornell | | Wisconsin | |
|---|---|---|---|---|---|---|
| Splits | 48/32/20(%)† | 60/20/20(%)‡ | 48/32/20(%)† | 60/20/20(%)‡ | 48/32/20(%)† | 60/20/20(%)‡ |
| GCN | 55.14 ± 5.16 | 79.33 ± 4.47 | 60.54 ± 5.30 | 69.53 ± 11.79 | 51.76 ± 3.06 | 63.94 ± 4.93 |
| GAT | 52.16 ± 6.63 | 79.59 ± 9.21 | 61.89 ± 5.05 | 66.91 ± 15.99 | 49.41 ± 4.09 | 63.45 ± 11.65 |
| GraphSAGE | 82.43 ± 6.14 | 86.02 ± 4.78 | 75.95 ± 5.01 | 85.06 ± 5.12 | 81.18 ± 5.56 | 89.56 ± 3.99 |
| MixHop | 77.84 ± 7.73 | − | 73.51 ± 6.34 | − | 75.88 ± 4.90 | − |
| Geom-GCN | 66.76 ± 2.72 | − | 60.54 ± 3.67 | − | 64.51 ± 3.66 | − |
| H$_2$GCN | 84.86 ± 7.23 | 85.90 ± 3.53 | 82.70 ± 5.28 | 86.23 ± 4.71 | 87.65 ± 4.98 | 87.50 ± 1.77 |
| LINKX | 74.60 ± 8.37 | − | 77.84 ± 5.81 | − | 75.49 ± 5.72 | − |
| WRGAT | 83.62 ± 5.50 | − | 81.62 ± 3.90 | − | 86.98 ± 3.78 | − |
| FAGCN | 82.43 ± 6.89 | 85.57 ± 4.75 | 79.19 ± 9.79 | 86.38 ± 5.33 | 82.94 ± 7.95 | 84.88 ± 9.19 |
| GPR-GNN | 78.38 ± 4.36 | 91.89 ± 4.08 | 80.27 ± 8.11 | 85.91 ± 4.60 | 82.94 ± 4.21 | 93.84 ± 3.16 |
| GGCN | 84.86 ± 4.55 | 92.13 ± 3.05 | 85.68 ± 6.63 | 88.70 ± 4.97 | 86.86 ± 3.29 | 94.56 ± 3.26 |
| NLMLP | 85.40 ± 3.80 | − | 84.90 ± 5.70 | − | 87.30 ± 4.30 | − |
| GloGNN∗ | 84.05 ± 4.90 | − | 85.95 ± 5.10 | − | 88.04 ± 3.22 | − |
| NSD∗ | 85.95 ± 5.51 | − | 84.86 ± 4.71 | − | 89.41 ± 4.74 | − |
| ACM-GCN∗ | 88.38 ± 3.43 | − | 86.49 ± 6.73 | − | 88.43 ± 3.66 | − |
| Ordered GNN | 86.22 ± 4.12 | 90.82 ± 4.18 | 87.03 ± 4.73 | 88.09 ± 3.36 | 88.04 ± 3.63 | 93.62 ± 2.91 |
| **GWN-sym** | 89.85 ± 5.05 | 91.64 ± 3.74 | 88.11 ± 3.17 | 89.57 ± 3.54 | 90.88 ± 4.43 | 93.38 ± 3.18 |
| **GWN-fa** | **92.94 ± 4.45** | **93.28 ± 3.14** | **90.81 ± 4.63** | **92.13 ± 3.76** | **94.26 ± 1.76** | **95.63 ± 1.59** |

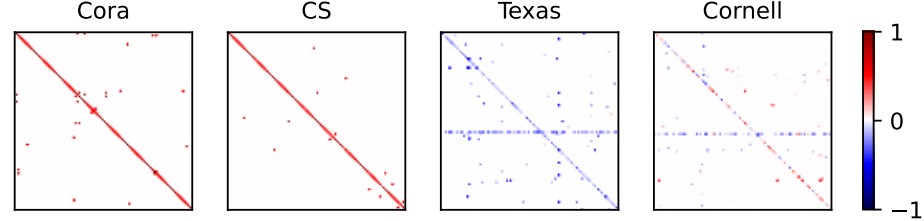

**Figure 2: Visualization of the attention matrix $\alpha$ of GWN-fa (selected from 100 nodes).**

indicates that *GWN outperforms heat equation based methods sush as GRAND on 5 datasets and achieves suboptimal performance on*

*Photo*. Since both GWN-sym and GWN-fa can be treated as low-pass filters, their performances are closely comparable. Then, we examine the generality of GWN on heterophilic datasets. Table 3



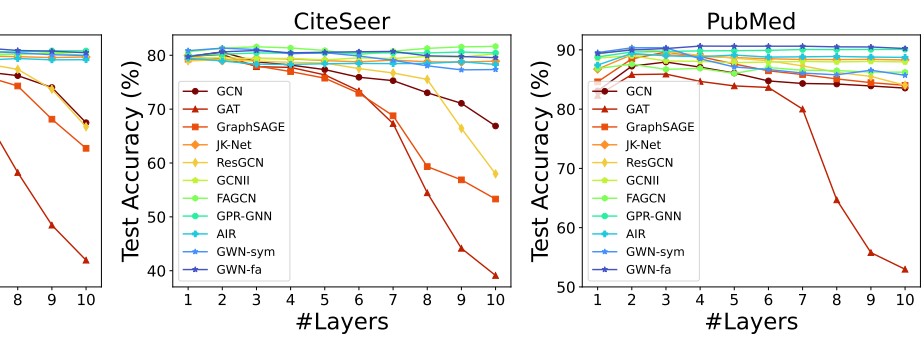

**Figure 3: Performances of methods of each layer on citation networks.**

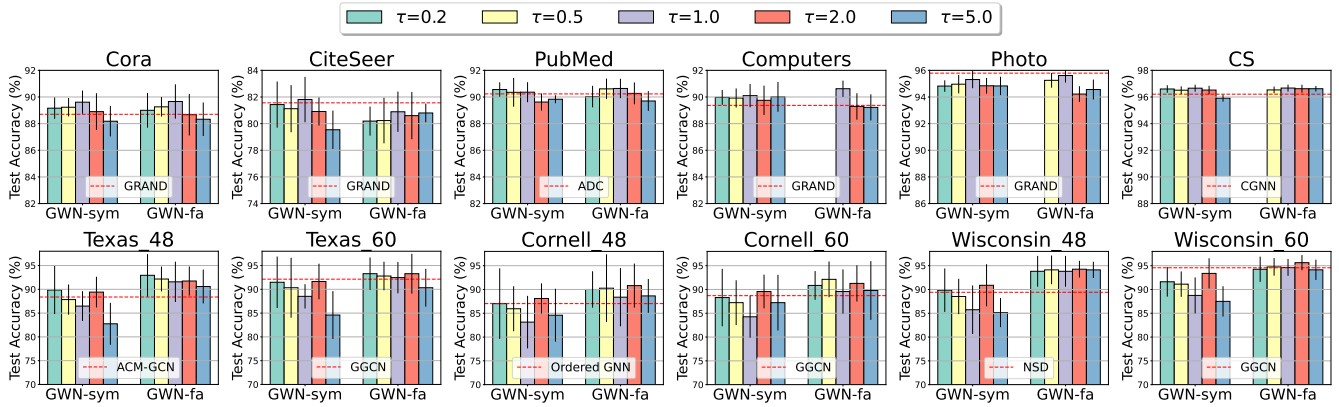

**Figure 4: Stability analysis of GWN. Red dashed line indicates the SOTA models.**

demonstrates that GWN outperforms state-of-the-art methods under two commonly used data splits. GWN-sym and GWN-fa show a significant difference on heterophilic graphs, which can be attributed to the high-pass filter of GWN-fa. *It reflects remarkable performance in modelling heterophily in graphs.* Next, we further probe whether the low-pass and high-pass filters in GWN-fa perform as expected. As shown in Figure 2, we visualize the attention matrix $\boldsymbol{\alpha}$ in Eq. (18). As stated in Sec. 4.4, GWN behaves as a low-pass filter when $\alpha_{ij} > 0$ and behaves as a high-pass filter when $\alpha_{ij} < 0$. This is consistent with the visualized results. Additionally, it is worth noting that under the 48%/32%/20% split, GWN-fa even outperforms most state-of-the-art methods under the 60%/20%/20% split, highlighting *its superiority in scenarios with low label rates.*

## 5.3 Over-smoothing Analysis

We investigate the ability of GWN to mitigate over-smoothing on three citation networks: Cora, CiteSeer, and PubMed. Following the practice of Chamberlain et al. [6] and Rusch et al. [34], we set $\tau = 1$, then the terminal time $T$ denotes the number of layers. Figure 3 demonstrates that compared to GNNs specifically designed for over-smoothing, GWN not only outperforms all baselines in terms of performance but also maintains its performance as the number of layers increases. It demonstrates that *our models have better performance in mitigating the over-smoothing issue.*

## 5.4 Stability Analysis

We analyze the stability of the explicit scheme in experiments. We run GWN on all datasets and report the accuracy for different $\tau$. As depicted in Figure 4, on larger-scale datasets such as CS, GWN exhibits minimal performance differences when $\tau$ is varied. On smaller-scale datasets like Texas, due to the high-pass filters, GWN-fa demonstrates better stability compared to GWN-sym. Compared to GRAND, which only guarantees stability of the explicit scheme for $\tau = 0.005$ [6], *GWN can maintain stability at larger $\tau$ while achieving higher computational efficiency and performance.*

## 5.5 Efficiency Analysis

We analyze the efficiency of GWN in Figure 5. We compare GWN at different $\tau$ with four basic GNNs and three variants of GRAND (GRAND-l, GRAND-nl, and GRAND-nl-rw). GWN-sym is faster than GWN-fa because it has fewer learnable parameters. And *as $\tau$ increases, their runtime becomes faster.* Taking CiteSeer as an example, "sym-1.0 (1.7x)" achieves both optimal performance and competitive efficiency. Furthermore, GRAND exhibits overall less efficiency, with its three variants mainly cluster on the right side of figures, and their runtimes are on the order of $10^1$. Considering its most efficient variant "l-0.5 (3.9x)", its performance and runtime are still inferior to our variant "sym-1.0 (1.7x)". It demonstrates that *our models can achieve both efficient and effective performance.*

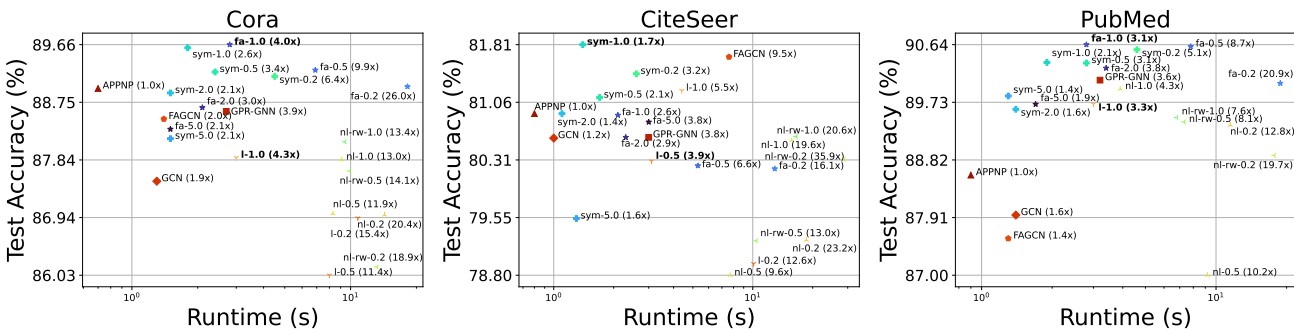

**Figure 5: Accuracy and runtime of the methods: "sym" and "fa" denote two variants of GWN; "l", "nl", and "nl-rw" denote three variants of GRAND; "-1.0" denotes $\tau = 1.0$; (2.0x) denotes the multiple of the shortest runtime.**

**Table 4: Comparison with base models using best parameters.**

|  | Cora | CiteSeer | PubMed | Computers | Photo | CS | Texas | Cornell | Wisconsin |
|---|---|---|---|---|---|---|---|---|---|
| GCN | 87.51 | 80.59 | 87.95 | 86.09 | 93.04 | 95.14 | 79.33 | 69.53 | 63.94 |
| **GWN-sym** | **89.61** (↑ 2.10) | **81.81** (↑ 1.22) | **90.56** (↑ 2.61) | **90.10** (↑ 4.01) | **95.31** (↑ 2.27) | **96.66** (↑ 1.52) | **91.64** (↑ 12.31) | **89.57** (↑ 20.04) | **93.38** (↑ 29.44) |
| FAGCN | 88.49 | **81.65** | 87.58 | 87.32 | 93.41 | 95.79 | 85.57 | 86.38 | 84.88 |
| **GWN-fa** | **89.66** (↑ 1.17) | 80.89 (↓ 0.76) | **90.64** (↑ 3.06) | **90.62** (↑ 3.30) | **95.61** (↑ 2.20) | **96.67** (↑ 0.88) | **93.28** (↑ 7.71) | **92.13** (↑ 5.75) | **95.63** (↑ 10.75) |

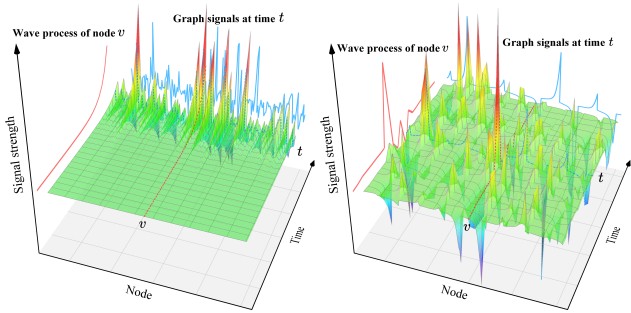

**Figure 6: Visualization of waveform in wave propagation of GWN-sym (Left) and GWN-fa (Right) on Cora.**

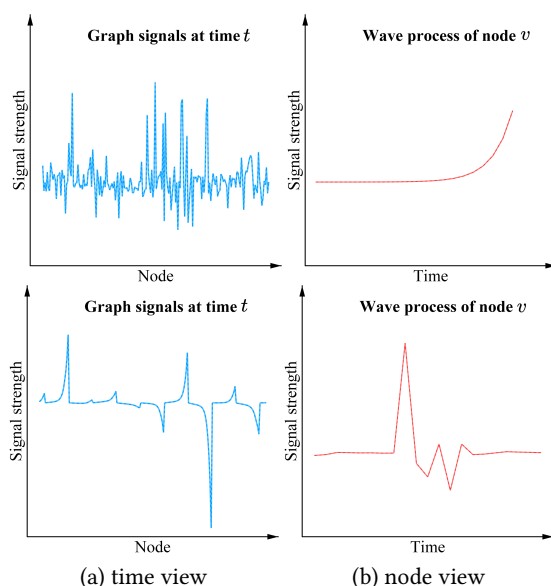

(a) time view    (b) node view

**Figure 7: Visualization of time and node view in wave propagation of GWN-sym (Up) and GWN-fa (Down) on Cora.**

## 5.6 Analysis of base models

We explore the impact of base models in Table 4, where both GCN and FAGCN increase their performances in the form of the graph wave equation. Besides, we also show the visualization of wave propagation of GWN-sym and GWN-fa on Cora. From Figure 6, the waveform of GWN-sym are not prominent at early iterations, while the waveform of GWN-fa demonstrate more frequent information interactions at any time. Figure 7(a) depicts wave signals at a specific time, where the waveform of GWN-sym appears chaotic, whereas the waveform of GWN-fa is relatively clear. From Figure 7(b), compared to the waveform of GWN-sym, the waveform of GWN-fa exhibits periodic variations of peaks and troughs. We believe that these phenomena can be attributed to the ability of capturing both the low- and high-frequency signals in GWN-fa.

## 6 CONCLUSION

In this paper, we consider the message passing in GNNs as a wave propagation process, and further develop graph wave networks

(GWNs) based on the proposed graph wave equation with spectral GNNs. We demonstrate that compared to the heat equation, the graph wave equation exhibits superior performance and stability. Extensive experiments demonstrate that our GWNs obtain accurate and efficient performance, and show effectiveness in addressing challenging graph problems such as over-smoothing and heterophily modelling. Our future work would explore more complex and general Laplacian polynomials to advance GNNs.

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

## A DETAILED MATHEMATICAL DERIVATION

### A.1 Proof of Proposition 1

Proof. Since $\mathbf{L} = \mathbf{D} - \mathbf{A}$, then

$\mathbf{L}\mathbf{X} = (\mathbf{D} - \mathbf{A})\mathbf{X}$

$$= \left( \begin{bmatrix} \sum_j A_{1j} & \cdots & 0 \\ \vdots & \ddots & \vdots \\ 0 & \cdots & \sum_j A_{Nj} \end{bmatrix} - \begin{bmatrix} A_{11} & \cdots & A_{1N} \\ \vdots & \ddots & \vdots \\ A_{N1} & \cdots & A_{NN} \end{bmatrix} \right) \begin{bmatrix} \mathbf{x}_1 \\ \vdots \\ \mathbf{x}_N \end{bmatrix}$$

$$= \begin{bmatrix} \sum_j A_{1j} - A_{11} & \cdots & -A_{1N} \\ \vdots & \ddots & \vdots \\ -A_{N1} & \cdots & \sum_j A_{Nj} - A_{N1} \end{bmatrix} \begin{bmatrix} \mathbf{x}_1 \\ \vdots \\ \mathbf{x}_N \end{bmatrix}$$

$$= \left[ \dots, \sum_{v_j \in \mathcal{N}(v_i)} (\mathbf{x}_i - \mathbf{x}_j), \dots \right]^\top. \tag{19}$$

where $A_{ij} = 1$, if $e_{ij} \in \mathcal{E}$; $A_{ij} = 0$, otherwise. Hence, the graph wave equation can be rewritten as:

$$\frac{\partial^2 \mathbf{X}}{\partial t^2} = a^2 \mathbf{L}\mathbf{X}. \tag{20}$$

$\square$

### A.2 Derivation of Explicit Scheme

For any node $v_i$ at time $t$, we first discretize $t$ as $t_n = n\tau (n = 0, 1, \dots)$. Then, considering its node representation $\mathbf{x}_i$ along with a time step $\tau$, its time difference of wave equation is given by:

$$\frac{\partial^2 \mathbf{X}(\mathbf{x}_i, t_n)}{\partial t^2} = \sum_{v_j \in \mathcal{N}(v_i)} L_{ij}^{(n)} \left( \mathbf{x}_j^{(n)} - \mathbf{x}_i^{(n)} \right), \tag{21}$$

where $L_{ij}^{(n)}$ denotes the element of $\mathbf{L}_a$. Moreover, we can expand at point $(\mathbf{x}_i, t_n)$ using Taylor series, and obtain

$$\frac{\mathbf{X}(\mathbf{x}_i, t_{n+1}) - 2\mathbf{X}(\mathbf{x}_i, t_n) + \mathbf{X}(\mathbf{x}_i, t_{n-1})}{\tau^2}$$
$$= \frac{\partial^2 \mathbf{X}(\mathbf{x}_i, t_n)}{\partial t^2} + \frac{\tau^2}{12} \frac{\partial^4 \mathbf{X}(\mathbf{x}_i, t_n)}{\partial t^4} + O(\tau^4). \tag{22}$$

By substituting Eq. (21) into Eq. (22), we obtain

$$\frac{\mathbf{X}(\mathbf{x}_i, t_{n+1}) - 2\mathbf{X}(\mathbf{x}_i, t_n) + \mathbf{X}(\mathbf{x}_i, t_{n-1})}{\tau^2}$$
$$= \sum_{v_j \in \mathcal{N}(v_i)} L_{ij}^{(n)} \left( \mathbf{x}_j^{(n)} - \mathbf{x}_i^{(n)} \right) + R_i^{(n)}(\mathbf{X}), \tag{23}$$

where $R_i^{(n)}(\mathbf{X}) = \frac{\tau^2}{12} \frac{\partial^4}{\partial t^4} \mathbf{X}(\mathbf{x}_i, t_n) + O(\tau^4)$ denotes the truncation error. By truncating it, we obtain the forward time difference

$$\frac{\mathbf{X}_i^{(n+1)} - 2\mathbf{X}_i^{(n)} + \mathbf{X}_i^{(n-1)}}{\tau^2} = \sum_{v_j \in \mathcal{N}(v_i)} L_{ij}^{(n)} \left( \mathbf{x}_j^{(n)} - \mathbf{x}_i^{(n)} \right), \tag{24}$$

where $\mathbf{X}_i^{(n)} (n = 1, 2, \dots)$ denotes the approximate value of $\mathbf{X}$ at $(\mathbf{x}_i, t_n)$. Transforming the Eq. (24) into matrix form

$$\frac{\mathbf{X}^{(n+1)} - 2\mathbf{X}^{(n)} + \mathbf{X}^{(n-1)}}{\tau^2} = \mathbf{L}_a^{(n)} \mathbf{X}^{(n)}. \tag{25}$$

We define the problem of solving the partial differential equation as an initial value problem, with the initial values given by the second-order central difference quotient:

$$\mathbf{X}^{(0)} = \varphi_0(\mathbf{X}), \tag{26}$$

$$\frac{\mathbf{X}^{(1)} - \mathbf{X}^{(-1)}}{2\tau} = \varphi_1(\mathbf{X}), \tag{27}$$

where $\varphi_1(\mathbf{X})$ and $\varphi_2(\mathbf{X})$ can be obtained through neural layers. Let $n = 0$, then

$$\frac{\mathbf{X}^{(1)} - 2\mathbf{X}^{(0)} + \mathbf{X}^{(-1)}}{\tau^2} = \mathbf{L}_a^{(0)} \mathbf{X}^{(0)}. \tag{28}$$

By eliminating $\mathbf{X}^{(-1)}$ using the initial value Eq. (27), we obtain the node representation at time $t_1$:

$$\mathbf{X}^{(1)} = \tau\varphi_1(\mathbf{X}) + \left( \mathbf{I} + \frac{\tau^2}{2} \mathbf{L}_a^{(0)} \right) \varphi_0(\mathbf{X}). \tag{29}$$

Finally, we can derive the node representation at time $t_{n+1}$:

$$\mathbf{X}^{(n+1)} = \left( 2\mathbf{I} + \tau^2 \mathbf{L}_a^{(n)} \right) \mathbf{X}^{(n)} - \mathbf{X}^{(n-1)}. \tag{30}$$

### A.3 Proof of Theorem 2

According to **Theorem 1.**, to prove the stability of the explicit scheme $\mathbf{X}^{(n+1)} = \mathbf{A}\mathbf{X}^{(n)}$, we would like to prove that $|\lambda|_{max} \leq 1$.

Proof. To facilitate the stability analysis of the explicit scheme $\mathbf{X}^{(n+1)} = \left( 2\mathbf{I} + \tau^2 \mathbf{L}_a \right) \mathbf{X}^{(n)} - \mathbf{X}^{(n-1)}$, we first need to transform it into the form of $\mathbf{U}^{(n+1)} = \mathbf{C}\mathbf{U}^{(n)}$. Therefore, we assume

$$\mathbf{U}^{(n+1)} = \begin{bmatrix} \mathbf{X}^{(n+1)} \\ \mathbf{X}^{(n)} \end{bmatrix}, \mathbf{C} = \begin{bmatrix} \mathbf{C}_1 & \mathbf{C}_2 \\ \mathbf{C}_3 & \mathbf{C}_4 \end{bmatrix}, \tag{31}$$

then, we can obtain the system of linear equations:

$$\begin{cases} \mathbf{X}^{(n+1)} = \mathbf{C}_1 \mathbf{X}^{(n)} + \mathbf{C}_2 \mathbf{X}^{(n-1)} \\ \mathbf{X}^{(n)} = \mathbf{C}_3 \mathbf{X}^{(n)} + \mathbf{C}_4 \mathbf{X}^{(n-1)} \end{cases}. \tag{32}$$

According the equation of the explicit scheme, we can obtain the values of each block matrix in $\mathbf{C}$:

$$\mathbf{C}_1 = 2\mathbf{I} + \tau^2 \mathbf{L}_a, \mathbf{C}_2 = -\mathbf{I}, \mathbf{C}_3 = \mathbf{I}, \mathbf{C}_4 = \mathbf{0}, \tag{33}$$

resulting in $\mathbf{C}^{(n)}$ as follows:

$$\mathbf{C} = \begin{bmatrix} 2\mathbf{I} + \tau^2 \mathbf{L}_a & -\mathbf{I} \\ \mathbf{I} & \mathbf{0} \end{bmatrix}. \tag{34}$$

According to the Lemma 1, now we only need to compute the spectral radius of matrix $\mathbf{C}$ to determine the condition that ensures the stability of the explicit scheme.

For convenience, we still use $\mathbf{C}_1$ instead of $2\mathbf{I} + \tau^2 \mathbf{L}_a$, and let $\lambda, \lambda'$ denote the eigenvalue of $\mathbf{C}, \mathbf{C}_1$, respectively. Then, the characteristic determinant of $\mathbf{C}$ is given by:

$$det\{\mathbf{C} - \lambda \mathbf{I}_{2N}\} = \begin{vmatrix} \mathbf{C}_1 - \lambda \mathbf{I} & -\mathbf{I} \\ \mathbf{I} & -\lambda \mathbf{I} \end{vmatrix} = \left| (1 + \lambda^2)\mathbf{I} - \lambda \mathbf{C}_1 \right|$$
$$= \left| \frac{1+\lambda^2}{\lambda} \mathbf{I} - \mathbf{C}_1 \right| = 0. \tag{35}$$

Evidently, $\left| \frac{1+\lambda^2}{\lambda} \mathbf{I} - \mathbf{C}_1 \right| = 0$ is the characteristic equation of $\mathbf{C}_1$, with the eigenvalue $\lambda' = \frac{1+\lambda^2}{\lambda}$. According to the $\mathbf{L}_a$'s eigenvalue belongs to the interval $[-1, 1]$, then $\lambda' \in [2 - \tau^2, 2 + \tau^2]$. Next, we will investigate the range of values for $\lambda$.

Considering the equation obtained from $\lambda' = \frac{1+\lambda^2}{\lambda}$:

$$\lambda^2 - \lambda'\lambda + 1 = 0, \ \lambda' \in [2 - \tau^2, 2 + \tau^2]. \tag{36}$$

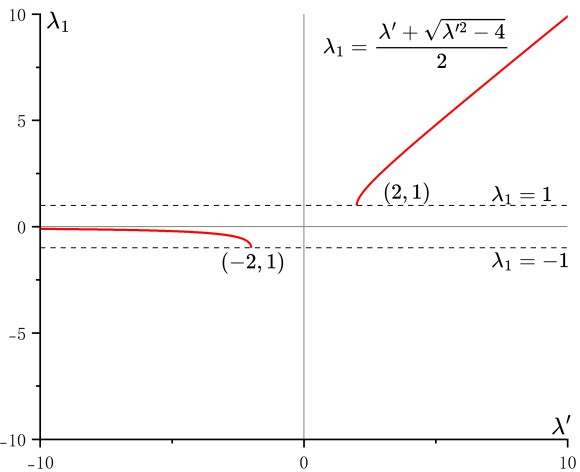
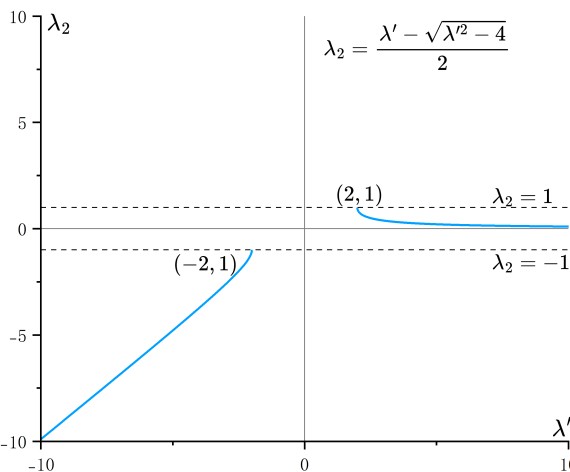

**Figure 8: The function curve of real roots $\lambda_1$ (Left) and $\lambda_2$ (Right).**

If $\Delta = \lambda'^2 - 4 < 0$, that is $-2 < \lambda' < 2$, then the equation has a pair of complex conjugate roots $\lambda_1 = \frac{\lambda' + i\sqrt{4-\lambda'^2}}{2}$ and $\lambda_2 = \frac{\lambda' - i\sqrt{4-\lambda'^2}}{2}$. And if $\Delta = \lambda'^2 - 4 \geq 0$, that is $|\lambda'| \geq 2$, then the equation has a pair of real roots $\lambda_1 = \frac{\lambda' + \sqrt{\lambda'^2-4}}{2}$ and $\lambda_2 = \frac{\lambda' - \sqrt{\lambda'^2-4}}{2}$. We next discuss whether the $|\lambda| \leq 1$ satisfies for different $\tau$, considering different cases.

**Case 1.** If $\tau \in (0, 2)$.

a) When $\lambda' \in [2 - \tau^2, 2)$, the equation has a pair of complex conjugate roots $\lambda_{1,2}$. Evidently, $|\lambda_{1,2}| = \sqrt{\left(\frac{\lambda'}{2}\right)^2 + \left(\frac{\sqrt{4-\lambda'^2}}{2}\right)^2} = 1$. Therefore, for any $\tau \in (0, 2)$, $|\lambda_{1,2}| \leq 1$.

b) When $\lambda' \in [2, 2 + \tau^2]$, the equation has a pair of real roots $\lambda_{1,2}$.

As shown in Figure 8 (Left), when $\lambda' = 2$, $|\lambda_1| = 1$, $\tau$ can take any positive real number at $(0, 2)$.

As shown in Figure 8 (Right), when $\lambda' \geq 2$, $|\lambda_2| \leq 1$, $\tau$ can take any positive real number at $(0, 2)$.

**Caes 2.** If $\tau \in [2, +\infty)$.

a) When $\lambda' \in [2 - \tau^2, -2]$, the equation has a pair of real roots $\lambda_{1,2}$.

As shown in Figure 8 (Left), when $\lambda' \leq -2$, $|\lambda_1| \leq 1$, $\tau$ can take any positive real number at $[2, +\infty)$.

As shown in Figure 8 (Right), when $\lambda' = -2$, $|\lambda_2| = 1$, $\tau$ can take any positive real number at $[2, +\infty)$.

b) When $\lambda' \in (-2, 2)$, the equation has a pair of complex conjugate roots $\lambda_{1,2}$. Evidently, $|\lambda_{1,2}| = 1$. Therefore, for any $\tau \in [2, +\infty)$, $|\lambda_{1,2}| \leq 1$.

c) When $\lambda' \in [2, 2 + \tau^2]$, the equation has a pair of real roots $\lambda_{1,2}$.

As shown in Figure 8 (Left), when $\lambda' = 2$, $|\lambda_1| = 1$, $\tau$ can take any positive real number at $[2, +\infty)$.

As shown in Figure 8 (Right), when $\lambda' \geq 2$, $|\lambda_2| \leq 1$, $\tau$ can take any positive real number at $[2, +\infty)$.

In conclusion, when $\tau \in R^+$, the eigenvalues $\lambda$ of $\mathbf{C}$ satisfy $|\lambda| \leq 1$, then the explicit scheme based on symmetric normalized Laplacian is constantly stable. □

### A.4 Derivation of GWN-fa

Given frequency adaptive Laplacian as follow:

$$\mathbf{L}_{a,l}^{(n)} = \varepsilon^{(n)}\mathbf{I} + \mathbf{D}^{-\frac{1}{2}}\mathbf{A}\mathbf{D}^{-\frac{1}{2}}, \quad \mathbf{L}_{a,h}^{(n)} = \varepsilon^{(n)}\mathbf{I} - \mathbf{D}^{-\frac{1}{2}}\mathbf{A}\mathbf{D}^{-\frac{1}{2}}, \quad (37)$$

where $\mathbf{L}_{a,l}^{(n)}$ is a low-pass filter, and $\mathbf{L}_{a,h}^{(n)}$ is a high-pass filter. Based on the aforementioned two Laplacian, we can capture the low-frequency and high-frequency signals at time $t_{n+1}$:

$$\mathbf{X}_l^{(n+1)} = \left(2\mathbf{I} + \tau^2\left(\varepsilon^{(n)}\mathbf{I} + \mathbf{D}^{-\frac{1}{2}}\mathbf{A}\mathbf{D}^{-\frac{1}{2}}\right)\right)\mathbf{X}^{(n)} - \mathbf{X}^{(n-1)},$$
$$\mathbf{X}_h^{(n+1)} = \left(2\mathbf{I} + \tau^2\left(\varepsilon^{(n)}\mathbf{I} - \mathbf{D}^{-\frac{1}{2}}\mathbf{A}\mathbf{D}^{-\frac{1}{2}}\right)\right)\mathbf{X}^{(n)} - \mathbf{X}^{(n-1)}. \quad (38)$$

Then, we adaptively combine low-frequency and high-frequency signals using attention weights and obtain the feature vector of node $v_i$ at time $t_{n+1}$:

$$\mathbf{x}_i^{(n+1)} = \alpha_{l,ij}^{(n)}\mathbf{x}_{l,i}^{(n+1)} + \alpha_{h,ij}^{(n)}\mathbf{x}_{h,i}^{(n+1)}$$

$$= \left(2 + \varepsilon^{(n)}\tau^2\right)\alpha_{l,ij}^{(n)}\mathbf{x}_i^{(n)} + \sum_{v_j \in \mathcal{N}(v_i)} \frac{\tau^2\alpha_{l,ij}^{(n)}}{\sqrt{deg(v_i)deg(v_j)}}\mathbf{x}_j^{(n)} - \alpha_{l,ij}^{(n)}\mathbf{x}_i^{(n-1)}$$

$$+ \left(2 + \varepsilon^{(n)}\tau^2\right)\alpha_{h,ij}^{(n)}\mathbf{x}_i^{(n)} - \sum_{v_j \in \mathcal{N}(v_i)} \frac{\tau^2\alpha_{h,ij}^{(n)}}{\sqrt{deg(v_i)deg(v_j)}}\mathbf{x}_j^{(n)} - \alpha_{h,ij}^{(n)}\mathbf{x}_i^{(n-1)}, \quad (39)$$

where $deg(v)$ denotes the degree of node $v$. Let $\alpha_{ij}^{(n)} = \alpha_{l,ij}^{(n)} - \alpha_{h,ij}^{(n)}$ and $\alpha_{l,ij}^{(n)} + \alpha_{h,ij}^{(n)} = 1$, we have

$$\mathbf{x}_i^{(n+1)} = \varepsilon^{(n)}\tau^2\mathbf{x}_i^{(n)}$$
$$+ \left(2\mathbf{x}_i^{(n)} + \sum_{v_j \in \mathcal{N}(v_i)} \frac{\tau^2\alpha_{ij}^{(n)}}{\sqrt{deg(v_i)deg(v_j)}}\mathbf{x}_j^{(n)}\right) - \mathbf{x}_i^{(n-1)}. \quad (40)$$

Weight $\alpha_{ij}^{(n)} = \tanh(\mathbf{g}^\top[\mathbf{x}_i^{(n)}||\mathbf{x}_j^{(n)}]) \in [-1, 1]$ can denote the correlation between nodes $v_i$ and $v_j$, and it is computed using attention mechanism, where $\mathbf{g} \in \mathbb{R}^{2d}$ is a learnable parameter vector. Since $\varepsilon$ is a learnable parameter, we can simplify $\varepsilon^{(n)}\tau^2$ as

$\varepsilon^{(n)}$. Furthermore, in order to preserve the original node features, we replace $\mathbf{x}_i^{(n)}$ with $\mathbf{x}_i^{(0)}$ [3]. So we can obtain

$$\mathbf{x}_i^{(n+1)} = \varepsilon^{(n)} \mathbf{x}_i^{(0)} + \left( 2\mathbf{x}_i^{(n)} + \sum_{v_j \in \mathcal{N}(v_i)} \frac{\tau^2 \alpha_{ij}^{(n)}}{\sqrt{deg(v_i)deg(v_j)}} \mathbf{x}_j^{(n)} \right) - \mathbf{x}_i^{(n-1)}. \tag{41}$$

Rewrite Eq. (41) in matrix form

$$\mathbf{X}^{(n+1)} = \varepsilon^{(n)} \mathbf{X}^{(0)} + \left( 2\mathbf{I} + \tau^2 \boldsymbol{\alpha}^{(n)} \odot \mathbf{D}^{-\frac{1}{2}} \mathbf{A} \mathbf{D}^{-\frac{1}{2}} \right) \mathbf{X}^{(n)} - \mathbf{X}^{(n-1)}, \tag{42}$$

where the $\alpha_{ij}^{(n)}$ is the element of $\boldsymbol{\alpha}^{(n)}$.

## A.5 Proof of Theorem 3

Similar to the proof of Theorem 2 (in A.3), we prove the stability of frequency adaptive Laplacian.

PROOF. Given $\mathbf{L}_a = \mathbf{D}^{-\frac{1}{2}} \mathbf{A} \mathbf{D}^{-\frac{1}{2}}$, we have

$$\mathbf{L}_{a,\cdot} = \varepsilon \mathbf{I} \pm \mathbf{D}^{-\frac{1}{2}} \mathbf{A} \mathbf{D}^{-\frac{1}{2}} = \varepsilon \mathbf{I} \pm \mathbf{L}_a, \tag{43}$$

and its eigenvalues satisfy $[\varepsilon - 1, \varepsilon + 1], \varepsilon \in (0, 1)$.

According to the proof of Theorem 1, when $\mathbf{L}_{a,\cdot}$'s eigenvalue belongs to the interval $[\varepsilon - 1, \varepsilon + 1]$, then the eigenvalue $\lambda'$ of $\mathbf{C}_1 = 2\mathbf{I} + \tau^2 \mathbf{L}_{a,\cdot}$ satisfies $\lambda' \in [2 + \tau^2(\varepsilon - 1), 2 + \tau^2(\varepsilon + 1)]$. Next, we will investigate the range of values for $\lambda$.

Considering the equation obtained from $\lambda' = \frac{1 + \lambda^2}{\lambda}$:

$$\lambda^2 - \lambda' \lambda + 1 = 0, \ \lambda' \in [2 + \tau^2(\varepsilon - 1), 2 + \tau^2(\varepsilon + 1)]. \tag{44}$$

If $\lambda'^2 - 4 < 0$, that is $-2 < \lambda' < 2$, then the equation has a pair of complex conjugate roots $\lambda_1 = \frac{\lambda' + i\sqrt{4 - \lambda'^2}}{2}$ and $\lambda_2 = \frac{\lambda' - i\sqrt{4 - \lambda'^2}}{2}$. And if $\lambda'^2 - 4 \geq 0$, that is $|\lambda'| \geq 2$, then the equation has real roots $\lambda_1 = \frac{\lambda' + \sqrt{\lambda'^2 - 4}}{2}$ and $\lambda_2 = \frac{\lambda' - \sqrt{\lambda'^2 - 4}}{2}$. We next discuss whether the $|\lambda| \leq 1$ satisfies for different $\tau$, considering different cases.

**Case 1.** If $\tau \in \left(0, \frac{2}{\sqrt{1-\varepsilon}}\right)$.

a) When $\lambda' \in [2 + \tau^2(\varepsilon - 1), 2)$, the equation has a pair of complex conjugate roots $\lambda_{1,2}$. Evidently, $|\lambda_{1,2}| = 1$. Therefore, for any $\tau \in \left(0, \frac{2}{\sqrt{1-\varepsilon}}\right)$, $|\lambda_{1,2}| \leq 1$.

b) When $\lambda' \in [2, 2 + \tau^2(\varepsilon + 1)]$, the equation has a pair of real roots $\lambda_{1,2}$.

As shown in Figure 8 (Left), when $\lambda' = 2$, $|\lambda_1| = 1$, $\tau$ can take any positive real number at $\left(0, \frac{2}{\sqrt{1-\varepsilon}}\right)$.

As shown in Figure 8 (Right), when $\lambda' \geq 2$, $|\lambda_2| \leq 1$, $\tau$ can take any positive real number at $\left(0, \frac{2}{\sqrt{1-\varepsilon}}\right)$.

**Caes 2.** If $\tau \in \left[\frac{2}{\sqrt{1-\varepsilon}}, +\infty\right)$.

a) When $\lambda' \in [2 + \tau^2(\varepsilon - 1), -2]$, the equation has a pair of real roots $\lambda_{1,2}$.

As shown in Figure 8 (Left), when $\lambda' \leq -2$, $|\lambda_1| \leq 1$, $\tau$ can take any positive real number at $\left[\frac{2}{\sqrt{1-\varepsilon}}, +\infty\right)$.

As shown in Figure 8 (Right), when $\lambda' = -2$, $|\lambda_2| = 1$, $\tau$ can take any positive real number at $\left[\frac{2}{\sqrt{1-\varepsilon}}, +\infty\right)$.

b) When $\lambda' \in (-2, 2)$, the equation has a pair of complex conjugate roots $\lambda_{1,2}$. Evidently, $|\lambda_{1,2}| = 1$. Therefore, for any $\tau \in \left[\frac{2}{\sqrt{1-\varepsilon}}, +\infty\right)$, $|\lambda_{1,2}| \leq 1$.

c) When $\lambda' \in [2, 2 + \tau^2(\varepsilon + 1)]$, the equation has a pair of real roots $\lambda_{1,2}$.

As shown in Figure 8 (Left), when $\lambda' = 2$, $|\lambda_1| = 1$, $\tau$ can take any positive real number at $\left[\frac{2}{\sqrt{1-\varepsilon}}, +\infty\right)$.

As shown in Figure 8 (Right), when $\lambda' \geq 2$, $|\lambda_2| \leq 1$, $\tau$ can take any positive real number at $\left[\frac{2}{\sqrt{1-\varepsilon}}, +\infty\right)$.

In conclusion, when $\tau \in R^+$, the eigenvalues $\lambda$ of $\mathbf{C}$ satisfy $|\lambda| \leq 1$, then the explicit scheme based on frequency adaptive Laplacian is constantly stable. □

## B IMPLEMENTATION DETAILS

For all experiments, each method was run on a single NVIDIA Tesla V100 GPU with 16GB memory, and the CPU used is Intel Xeon E5-2660 v4 CPUs. All models train 200 epochs and employed an early stopping strategy triggered when the loss exceed the average loss of the last 10 epochs.

### B.1 Dataset Statistics

As shown in Table 5, we report the statistical information of datasets used in this paper.

**Table 5: Dataset statistics.**

| Datesets | #Nodes | #Edges | #Features | #Classes |
| --- | --- | --- | --- | --- |
| Cora | 2708 | 5278 | 1433 | 7 |
| CiteSeer | 3327 | 4552 | 3703 | 6 |
| PubMed | 19717 | 44324 | 500 | 3 |
| Computers | 13752 | 245861 | 767 | 10 |
| Photo | 7650 | 119081 | 745 | 8 |
| CS | 18333 | 81894 | 6805 | 15 |
| Texas | 183 | 325 | 1703 | 5 |
| Cornell | 183 | 298 | 1703 | 5 |
| Wisconsin | 251 | 515 | 1703 | 5 |

### B.2 Parameter Search

We employ the wandb library for parameter search with Bayes scheme. The ranges for each hyperparameter are outlined in Table 6.

## C ADDITIONAL EXPERIMENTAL RESULTS

### C.1 Performance and Efficiency

As shown in Table 7 and Table 8, we present the complete performance and efficiency of GWN at different $\tau$. The results and runtimes for all models are obtained using the optimal parameters. It is important to note that the runtime, in addition to the time step $\tau$, can be influenced by parameters such as the number of layers, learning rate, and hidden layer dimensions. Therefore, there may be cases where the runtime increases for larger $\tau$ compared to smaller $\tau$.

Table 6: The range of hyperparameters.

| Hyperparameters | Range | Distribution |
|---|---|---|
| The dimension of hidden layer $d$ | {32, 64, 128, 256} | set |
| Time size $\tau$ | {0.2, 0.5, 1.0, 2.0, 5.0} | set |
| Terminal time $T$ | [1, 20] | uniform |
| Dropout | [0, 0.8] | uniform |
| Learning rate | [0.001, 0.25] | log_uniform_values |
| Weight decay | [0, 0.1] | uniform |

Table 7: The performances (%) and runtimes (s) of methods on homophilic datasets: mean accuracy ± standard deviation on 60%/20%/20% random splits and 10 runs. sym-0.2 indicates GWN-sym under $\tau = 0.2$. **Bold** and underline indicate optimal and suboptimal results, respectively. OOM denotes out of memory.

| Datesets | Cora | CiteSeer | PubMed | Computers | Photo | CS |
|---|---|---|---|---|---|---|
| SGC | 86.10 ± 1.37 (0.7) | 80.35 ± 1.33 (0.7) | 83.45 ± 0.51 (0.8) | 84.45 ± 0.56 (0.7) | 88.95 ± 0.86 (0.6) | 95.15 ± 0.24 (0.7) |
| APPNP | 88.97 ± 0.88 (0.7) | 80.91 ± 1.43 (0.8) | 88.58 ± 0.56 (0.9) | 86.73 ± 0.74 (0.8) | 93.74 ± 0.57 (0.8) | 95.58 ± 0.20 (1.6) |
| GPR-GNN | 88.61 ± 1.28 (2.7) | 80.60 ± 1.27 (3.0) | 90.08 ± 0.70 (3.2) | 88.55 ± 0.90 (1.4) | 94.61 ± 0.68 (2.5) | 96.26 ± 0.27 (1.3) |
| GCN | 87.51 ± 1.38 (1.3) | 80.59 ± 1.12 (1.0) | 87.95 ± 0.83 (1.4) | 86.09 ± 0.61 (1.3) | 93.04 ± 0.53 (1.4) | 95.14 ± 0.25 (0.9) |
| FAGCN | 88.49 ± 1.18 (1.4) | 81.65 ± 0.96 (7.6) | 87.58 ± 1.15 (1.3) | 87.32 ± 0.51 (1.2) | 93.41 ± 0.76 (1.0) | 95.79 ± 0.26 (4.8) |
| sym-0.2 | 89.16 ± 0.80 (4.5) | 81.43 ± 1.73 (2.6) | **90.56 ± 0.54** (4.6) | 89.97 ± 0.56 (8.8) | 94.82 ± 0.43 (4.1) | 96.60 ± 0.28 (3.6) |
| sym-0.5 | 89.23 ± 0.69 (2.4) | 81.12 ± 1.77 (1.7) | 90.35 ± 1.08 (2.8) | 89.92 ± 0.72 (3.8) | 94.96 ± 0.69 (3.6) | 96.51 ± 0.29 (2.8) |
| sym-1.0 | **89.61 ± 0.87** (1.8) | **81.81 ± 1.70** (1.4) | 90.36 ± 0.75 (1.9) | **90.10 ± 0.87** (3.7) | **95.31 ± 0.65** (4.0) | **96.66 ± 0.26** (3.2) |
| sym-2.0 | 88.90 ± 1.38 (1.5) | 80.91 ± 1.05 (1.1) | 89.62 ± 0.64 (1.4) | 89.75 ± 1.10 (2.0) | 94.84 ± 0.59 (1.9) | 96.52 ± 0.32 (2.3) |
| sym-5.0 | 88.18 ± 1.14 (1.5) | 79.54 ± 1.44 (1.4) | 89.83 ± 0.30 (1.3) | 90.01 ± 1.12 (2.9) | 94.83 ± 0.69 (1.4) | 95.91 ± 0.26 (3.1) |
| fa-0.2 | 89.00 ± 1.30 (18.2) | 80.19 ± 1.09 (12.9) | 90.03 ± 0.81 (18.8) | OOM | OOM | OOM |
| fa-0.5 | 89.26 ± 0.73 (6.9) | 80.23 ± 1.71 (5.3) | 90.61 ± 0.77 (7.8) | OOM | 95.25 ± 0.55 (6.9) | 96.53 ± 0.25 (5.5) |
| fa-1.0 | **89.66 ± 1.29** (2.8) | **80.89 ± 1.51** (2.1) | **90.64 ± 0.73** (2.8) | **90.62 ± 0.61** (5.6) | **95.61 ± 0.53** (3.5) | **96.67 ± 0.26** (8.6) |
| fa-2.0 | 88.67 ± 1.56 (2.1) | 80.60 ± 1.77 (2.3) | 90.27 ± 0.82 (3.4) | 89.29 ± 0.99 (5.9) | 94.22 ± 0.59 (1.7) | 96.62 ± 0.30 (2.5) |
| fa-5.0 | 88.33 ± 1.26 (1.5) | 80.80 ± 0.70 (3.0) | 89.70 ± 0.74 (1.7) | 89.22 ± 0.98 (3.6) | 94.56 ± 0.75 (2.3) | 96.62 ± 0.21 (5.4) |

Table 8: The performances (%) and runtimes (s) of methods on heterophilic datasets: mean accuracy ± standard deviation on random splits and 10 runs. sym-0.2 indicates GWN-sym under $\tau = 0.2$. **Bold** and underline indicate optimal and suboptimal results, respectively.

| Datesets | Texas | | Cornell | | Wisconsin | |
|---|---|---|---|---|---|---|
| Splits | 48/32/20(%) | 60/20/20(%) | 48/32/20(%) | 60/20/20(%) | 48/32/20(%) | 60/20/20(%) |
| SGC | - | 74.90 ± 8.23 (0.6) | - | 60.60 ± 11.92 (0.6) | - | 63.75 ± 5.14 (0.8) |
| APPNP | - | 81.64 ± 3.17 (1.2) | - | 73.40 ± 4.62 (1.3) | - | 71.50 ± 5.92 (1.5) |
| GPR-GNN | - | 91.89 ± 4.08 (0.9) | - | 85.91 ± 4.60 (0.8) | - | 93.84 ± 3.16 (0.9) |
| GCN | - | 79.33 ± 4.47 (2.2) | - | 69.53 ± 11.79 (2.9) | - | 63.94 ± 4.93 (1.1) |
| FAGCN | - | 85.57 ± 4.75 (4.6) | - | 86.38 ± 5.33 (3.7) | - | 84.88 ± 9.19 (5.6) |
| sym-0.2 | **89.85 ± 5.05** (2.3) | 91.48 ± 5.40 (2.6) | 87.03 ± 7.41 (2.2) | 88.30 ± 6.04 (2.7) | 89.85 ± 4.57 (2.2) | 91.62 ± 3.12 (2.5) |
| sym-0.5 | 87.84 ± 3.18 (1.7) | 90.33 ± 6.30 (1.6) | 85.95 ± 4.73 (1.8) | 87.23 ± 4.70 (1.7) | 88.53 ± 3.72 (1.7) | 91.12 ± 2.67 (1.7) |
| sym-1.0 | 86.47 ± 3.13 (1.4) | 88.52 ± 2.56 (1.4) | 83.14 ± 5.52 (1.4) | 84.26 ± 4.40 (1.4) | 85.74 ± 5.10 (1.4) | 88.75 ± 3.78 (1.5) |
| sym-2.0 | 89.41 ± 3.23 (1.1) | **91.64 ± 3.74** (1.2) | **88.11 ± 3.17** (1.1) | **89.57 ± 3.54** (1.1) | **90.88 ± 4.43** (1.0) | **93.38 ± 3.18** (1.1) |
| sym-5.0 | 82.75 ± 4.41 (1.1) | 84.59 ± 5.02 (1.2) | 84.59 ± 5.56 (1.1) | 87.23 ± 5.85 (1.1) | 85.15 ± 3.06 (1.2) | 87.50 ± 4.49 (1.2) |
| fa-0.2 | **92.94 ± 4.45** (4.4) | **93.28 ± 3.41** (5.1) | 90.00 ± 3.83 (4.2) | 90.85 ± 3.02 (5.9) | 93.82 ± 3.24 (3.7) | 94.25 ± 2.65 (4.2) |
| fa-0.5 | 92.16 ± 2.61 (2.1) | 92.79 ± 3.01 (2.4) | 90.27 ± 7.12 (2.1) | **92.13 ± 3.76** (2.0) | 94.12 ± 3.02 (2.0) | 94.75 ± 1.85 (2.1) |
| fa-1.0 | 91.57 ± 4.24 (1.8) | 92.46 ± 3.30 (1.8) | 88.38 ± 6.12 (1.8) | 89.57 ± 4.65 (1.7) | 93.82 ± 3.24 (2.0) | 94.63 ± 1.77 (1.8) |
| fa-2.0 | 91.74 ± 3.06 (1.4) | 93.28 ± 4.19 (1.4) | **90.81 ± 4.63** (1.4) | 91.28 ± 3.81 (1.7) | **94.26 ± 1.76** (1.3) | **95.63 ± 1.59** (1.3) |
| fa-5.0 | 90.59 ± 3.56 (1.3) | 90.33 ± 3.97 (1.3) | 88.65 ± 3.56 (1.3) | 89.79 ± 6.17 (1.5) | 94.12 ± 1.70 (1.2) | 94.13 ± 2.13 (1.4) |

