# OpenReview forum: "Graph Wave Networks"
_ACM.org/TheWebConf/2025/Conference — WWW 2025 Oral_

### Official Review · Reviewer_pnBD · 2024-11-08

**Novelty:** 6
**Technical Quality:** 6

**Review:**

The paper introduces a novel approach to dynamic Graph Neural Networks (GNNs) by leveraging the graph wave equation instead of the commonly used heat diffusion process for message-passing mechanisms.
The authors argue that current dynamic GNNs based on heat diffusion processes are limited by their inability to capture the wave-like propagation characteristics often inherent in graph structures. To address this limitation, the paper proposes a graph wave equation-based model that improves performance and demonstrates enhanced temporal stability. The shift from a first-order partial differential equation (PDF), as in heat diffusion, to a second-order PDF in the wave equation provides a more robust framework for modeling dynamic message propagation theoretically and empirically.
The strengths of this paper are 1) Logical thinking and clear structure. The paper is logically structured, with each section building effectively to identify the shortcomings of the heat diffusion process in representing dynamic graph message-passing and explain the benefit of adopting a wave equation approach. 2) Concise and precise language, avoiding redundant repetition. 3) comprehensive literature review, drawing on a broad range of existing work in dynamic GNNs and PDF-based methods. 4) Mathematical rigor, sufficient reasoning, and thorough validation. By basing the model on a second-order PDE (the wave equation), the authors provide a mathematically sound justification for the stability advantages of their model. This stability in the time dimension is a notable improvement over the previous first-order heat diffusion models, which often encounter issues with long-term temporal consistency. The clear and complete presentation of mathematical proofs adds credibility to their claims. 5) Enhanced performance and stability. 6) Elegant writing and presentation.

**Questions:**

This is a highly-qualified paper, though two minor flaws could be refined.
1) The paper does a commendable job in Section 4.1 for connecting the graph to wave-like behavior. However, these connections are already well-established by the spectral GCNs through the graph Laplacian and the Graph Fourier. I think there appears to be some overlap or repetition in the discussion of this content between Section 4.2 and the "Spectral Graph Convolution" part in Section 3.1.
2) In Appendix A.1, there's a minor symbol error in the proof. I think the correct expression should be $$A_{Nj}-A_{NN}$$ rather than $$A_{Nj}-A_{N1}$$ in the lower right corner..
3) To further demonstrate the generalization and applicability of your Graph Wave Networks, please consider adding experiments on the OGB (Open Graph Benchmark) dataset to enhance clarity and facilitate understanding and review by others.

**Reviewer Confidence:**

3: The reviewer is confident but not certain that the evaluation is correct

**Scope:**

4: The work is relevant to the Web and to the track, and is of broad interest to the community

---

### Official Review · Reviewer_Y7Mw · 2024-11-29

**Novelty:** 4
**Technical Quality:** 3

**Review:**

**Summary**

In this paper, the authors revisit the message-passing mechanism of graph neural networks. They argue that in graph signal processing, the heat equation struggles to capture the wave-like properties of graph signals. The heat equation is essentially a partial differential equation involving a first-order time derivative, which has low numerical stability and leads to inefficient model training. To address this, the authors treat message passing (MP) as a wave propagation process, aiming to capture the temporal evolution of wave signals in space. Based on the wave equation from physics, the paper establishes a graph wave equation to leverage wave propagation on graphs. Furthermore, the authors demonstrate that the graph wave equation can be connected to traditional spectral GNNs, enabling the design of graph wave networks (GWNs) based on various Laplacians, which improves the performance of spectral GNNs. In addition, the authors argue that the proposed GWNs offers greater stability on graphs compared to the heat equation, which involves a first-order time derivative. This enhanced stability allows the model to significantly improve efficiency while maintaining high performance. The authors demonstrate the superior performance of the proposed model on several datasets.

**Strengths**

1. This article is well-structed and easy to follow.

2. This article is well-motivated and clearly presents the underlying principles behind the designed model.

3. The effectiveness of the proposed method is demonstrated on the evaluation datasets. Moreover, the authors conduct extensive experiments to verify its good properties.

**Weaknesses**
1. My major concern is lies in the experiments. The datasets used by the authors, whether homophily (e.g., Cora and PubMed) or heterophily (e.g., Texas and Cornell), can be considered "toy datasets." These datasets have a maximum of only around 20,000 nodes, which is far from representative of real-world graphs. While the proposed model shows good performance on these datasets, how does it perform on larger-scale datasets? There are already many large-scale graph datasets available, such as the homophily graphs like ogbn-arxiv [1] and ogbn-products [1], and heterophily graphs like Amazon-ratings [2], Roman-empire [2], and Questions [2]. Why isn't the proposed model tested on these datasets to validate its effectiveness? This would further strengthen the demonstration of the effectiveness of GWNs and showcase its scalability. If the model only performs well on such small datasets, it would limit the range of real-world scenarios where it could be applied.

[1] Open graph benchmark: Datasets for machine learning on graphs. In NeurIPS.

[2] A Critical Look at the Evaluation of GNNs under Heterophily: Are We Really Making Progress? In ICLR.

2. The heterophily datasets used in the paper have been shown in [2] to suffer from a serious issue of node duplication, which could undermine the credibility of the proposed model's claimed superiority.

3. The baselines compared by the authors are quite outdated. It would be more convincing to compare with some recently proposed models, as this would further strengthen the persuasiveness of the authors' claims.

**Questions:**

Please address the raised issues as thoroughly as possible in the weaknesses section.

**Reviewer Confidence:**

3: The reviewer is confident but not certain that the evaluation is correct

**Scope:**

3: The work is somewhat relevant to the Web and to the track, and is of narrow interest to a sub-community

---

### Official Review · Reviewer_gDRu · 2024-11-29

**Novelty:** 6
**Technical Quality:** 6

**Review:**

**Quality**

The paper introduces Graph Wave Networks, a novel method that improves the message passing paradigm in Graph Neural Networks. By extending the classical wave equation to graph domain, the method models message passing as a wave propagation process. The quality of the paper is high, supported by a solid theoretical foundation and extensive experimental validation.

**Clarity**

The paper is well-organized and the numerous tables and figures demonstrate the method's efficacy. However, the mathematical formulations might be challenging for readers without a strong background in graph signal processing or partial differential equations. Some parts could benefit from more intuitive explanations or simplified derivations to increase accessibility.

**Originality**

The paper innovatively models message passing in GNNs as a wave propagation process. This new perspective provides a fresh approach to understanding node interactions in graphs, making the contribution original.

**Significance**

This method addresses long-standing issues in GNNs, such as over-smoothing and heterophily. Both theoretical and empirical results demonstrate that it enables more efficient and robust graph-based learning.

**Pros**

1. The introduction of wave-based message passing is a novel contribution that expands the theoretical framework of GNNs and offers a fresh perspective on graph signal processing.
2. The paper provides rigorous theoretical proofs and extensive experiments across several benchmark datasets, thoroughly validating the proposed method.
3. The method effectively addresses long-standing challenges in GNNs, such as over-smoothing and heterogeneity, achieving more efficient and robust graph-based learning.

**cons**

1. Although GWN performs excellently on small-scale datasets, its computational complexity remains high, which could potentially pose challenges in terms of resource requirements and scalability.
2. The introduction of wave equations adds significant mathematical complexity. More intuitive explanations would improve accessibility for a broader audience.
3. Although the proposed method is innovative, it still relies on the traditional spectral GNN framework, which may limit its potential for further innovation and flexibility.

**Questions:**

1. The paper demonstrates excellent performance on benchmark datasets, but how well does the GWNs scale to larger graphs with millions of nodes and edges?
2. How interpretable is the wave-based message-passing mechanism in practice?
3. The paper claims that GWNs mitigate the over-smoothing problem. Could you elaborate from a theoretical perspective on what makes GWNs particularly effective for this challenge?

**Reviewer Confidence:**

3: The reviewer is confident but not certain that the evaluation is correct

**Scope:**

4: The work is relevant to the Web and to the track, and is of broad interest to the community

---

### Official Review · Reviewer_BtFo · 2024-11-30

**Novelty:** 6
**Technical Quality:** 6

**Review:**

This paper introduces Graph Wave Networks (GWNs), a novel approach to message passing in Graph Neural Networks (GNNs) inspired by wave propagation and grounded in the wave equation from physics. It provides a rigorous theoretical framework and extensive empirical validation, showcasing robustness and efficiency compared to state-of-the-art methods. The paper is well-structured and contributes significantly to advancing GNN design, particularly in addressing challenges such as over-smoothing and heterophily modeling.

Pros:

1, Innovative Perspective: GWNs redefine message passing by modeling it as a wave propagation process, leveraging the wave equation rather than the conventional heat equation. This paradigm shift is both creative and practical, effectively addressing limitations of existing models.

2, Theoretical Rigor: The paper establishes a strong theoretical foundation for the Graph Wave Equation, including stability proofs and connections to spectral GNNs. These insights provide a deeper understanding of wave-based message passing and its potential benefits.
3, Empirical Excellence: Extensive experiments demonstrate superior performance across diverse datasets, effectively handling both homophilic and heterophilic graph settings. The results are robust, supported by comparisons with numerous baselines, comprehensive ablation studies, and efficiency analyses.

Cons:

1,  The assumption that the wave nature of graph signals applies universally to all graph types may not hold for highly dynamic or irregular graphs. Providing additional empirical evidence to support this assumption would strengthen the paper’s claims.

2,  The use of the forward Euler method to solve the wave equation is practical, but further analysis of its limitations and potential alternatives could enhance the argument for its adoption.

3, While the spectral graph convolution adaptations are well-executed, explicitly linking these adaptations to specific gaps in prior models would more clearly demonstrate the paper’s innovation.

4, The limited exploration of individual components, such as the contributions of low-pass versus high-pass filters, leaves gaps in understanding their relative impact. More detailed ablation experiments would clarify their roles and improve interpretability.

**Questions:**

See Cons.

**Reviewer Confidence:**

3: The reviewer is confident but not certain that the evaluation is correct

**Scope:**

4: The work is relevant to the Web and to the track, and is of broad interest to the community